# Increasing Adult Hippocampal Neurogenesis Promotes Resilience in a Mouse Model of Depression

**DOI:** 10.3390/cells10050972

**Published:** 2021-04-21

**Authors:** Barbara Planchez, Natalia Lagunas, Anne-Marie Le Guisquet, Marc Legrand, Alexandre Surget, René Hen, Catherine Belzung

**Affiliations:** 1UMR 1253, iBrain, Université de Tours, Inserm, CEDEX 1, 37032 Tours, France; barbara.planchez@etu.univ-tours.fr (B.P.); llagunas@ucm.es (N.L.); le.guisquet@univ-tours.fr (A.-M.L.G.); marc.legrand@etu.univ-tours.fr (M.L.); alexandre.surget@univ-tours.fr (A.S.); 2Departments of Neuroscience, Psychiatry & Pharmacology, Columbia University, New York, NY 10027, USA; rh95@cumc.columbia.edu; 3Division of Integrative Neuroscience, Department of Psychiatry, New York State Psychiatric Institute, New York, NY 10032, USA; 4Kavli Institute for Brain Sciences, Columbia University, New York, NY 10027, USA

**Keywords:** depression, adult hippocampal neurogenesis, chronic stress, inhibition, flexibility, depressive-like behaviors, hippocampus, stress resilience

## Abstract

Many studies evaluated the functional role of adult hippocampal neurogenesis (AHN) and its key role in cognitive functions and mood regulation. The effects of promoting AHN on the recovery of stress-induced symptoms have been well studied, but its involvement in stress resilience remains elusive. We used a mouse model enabling us to foster AHN before the exposure to unpredictable chronic mild stress (UCMS) to evaluate the potential protective effects of AHN on stress, assessing the depressive-like phenotype and executive functions. For this purpose, an inducible transgenic mouse model was used to delete the pro-apoptotic gene *Bax* from neural progenitors four weeks before UCMS, whereby increasing the survival of adult-generated neurons. Our results showed that UCMS elicited a depressive-like phenotype, highlighted by a deteriorated coat state, a higher immobility duration in the tail suspension test (TST), and a delayed reversal learning in a water maze procedure. Promoting AHN before UCMS was sufficient to prevent the development of stressed-induced behavioral changes in the TST and the water maze, reflecting an effect of AHN on stress resilience. Taken together, our data suggest that increasing AHN promotes stress resilience on some depressive-like symptoms but also in cognitive symptoms, which are often observed in MD.

## 1. Introduction

Major depression (MD) is a highly debilitating disorder that affects millions of people worldwide and is thus a main contributor to the global burden of healthcare and the economy. Although many pharmacological approaches have been developed to alleviate symptoms, almost a third of patients still exhibit symptoms after receiving a treatment [1,2], and only 30% go through remission. Even though MD is not a homogenous pathology, it exhibits core symptoms such as anhedonia and depressive mood, often associated with sleep and eating disturbances, psychomotor alterations, and anxiety.

Antidepressants currently in use are mainly based on active molecules modulating monoamine systems; among classic drugs are the selective serotonin reuptake inhibitors (SSRIs). The efficacy of these drugs appears after a delay, leading to the idea that their antidepressant effects are not solely due to the direct modulation of monoamines but are also the consequence of long-term downstream changes. As a matter of fact, antidepressants induce therapeutic effects along with an increase in hippocampal neurogenesis [3,4,5,6,7,8,9]. Interestingly, rather than being a concomitant process, the increase in adult hippocampal neurogenesis (AHN) would be causal because its inhibition or depletion blocks some effects induced by antidepressants [10,11,12,13,14,15].

AHN is a complex process by which adult-born neurons are continuously generated in the hippocampus during adulthood in mammals. Neuronal stem cells, located in the subgranular zone of the dentate gyrus (DG) of the hippocampus, go through successive steps of proliferation, migration, and maturation in order to develop into neurons fully incorporated into pre-existing circuits. The role of adult-born neurons has been extensively studied since their discovery by Altman and colleagues [16]. Interestingly, AHN is implicated in cognitive functions such as memory and learning, as well as in mood and stress regulation [11,17,18,19,20,21,22,23,24]. While precise mechanisms remain elusive, recent studies support the idea that adult-born neurons could act as key regulators of global hippocampal activity. Indeed, while inhibition of AHN can increase the overall excitability of the hippocampus, stimulation or promotion of AHN lowers mature granule cells’ activity in the DG [25,26,27]. Thereby, AHN could participate in functions supported directly by the hippocampus. Additionally, very recent studies showed that enhancement of AHN could not only affect neuronal activity in trisynaptic hippocampal networks but also in downstream circuits [28,29,30].

Many factors can act on AHN, such as enrichment and antidepressants, which positively regulate it [31,32], while stress negatively regulates it by decreasing the proliferation, maturation, and survival of adult-born neurons [33,34,35]. Because stress is considered as a trigger for MD [36] and that antidepressants induce behavioral responses supported by AHN [11,12,13,14,37], it is supposed that a stress-induced impairment of AHN could be a key factor in the pathophysiology of stress-related disorders such as MD. Accordingly, many studies focused on the antidepressant effects of AHN on stress-induced impairments once the symptoms were present, mainly reflecting the effects of AHN on the recovery, while the protective role of AHN remains less studied. However, the involvement of AHN in stress resilience is supported by the fact that factors that positively regulate AHN tend to also promote resilience [38]. Additionally, because AHN buffers stress responses [39], it is likely that increasing AHN could prevent the deleterious effects of stress on cognition and mood. Indeed, some recent papers highlighted a potential role of AHN in the resilience to stress: while the inhibition of the activity of adult-born neurons can lead to a vulnerability to social defeat, increasing AHN promotes resilience [27]. Similarly, in an animal model of chronic stress based on chronic corticosterone injections, increasing AHN was able to protect preventively from stress-induced depressive-like behaviors [18].

Given the functional role of adult-born neurons, promoting AHN before stress could prevent stress-induced impairments on hippocampus-dependent functions, which are usually impaired in MD. Furthermore, if cognitive alterations are not the core symptoms of MD, it is currently well known that patients suffering from MD exhibit memory deficits, decreased flexibility, and impaired inhibitory control [40], which could then be prevented by increasing AHN. Nonetheless, studies that tried to highlight a link between AHN and stress resilience focused on anxiety and depressive-like behaviors while the effects on cognitive functions are still poorly assessed. Additionally, because the hippocampus is part of the corticolimbic system, normalizing its activity through AHN enhancement could also affect interconnected regions involved in stress and mood regulation [41,42] and thereby prevent the development of depressive and anxiety-like behaviors.

In an attempt to shed more light on the potential link between AHN and stress resilience, we stimulated AHN in an animal model of unpredictable chronic mild stress (UCMS), which is a naturalistic model of MD [43,44]. We used an inducible transgenic mouse model in which it is possible to remove the pro-apoptotic Bax gene from neural stem cells at the time of tamoxifen injection, which in turn promotes adult-generated neuron survival and increases AHN [17]. In order to quantify the increase in AHN, survival and maturation indexes were assessed by using doublecortin immunochemistry, a marker of immature neurons. We first analyzed the effect of chronic stress and AHN on anxiety and depressive-like behaviors; then, because cognitive deficits are often observed in MD [40], we evaluated through a water maze paradigm the functions of flexibility and inhibition in UCMS mice and the effects of increasing AHN on them.

## 2. Method

### 2.1. Animals

Male iBax mice aged 11 weeks at the start of the experiments were used (*N* = 83). The generation and characterization of the *Nes-CreER^T2^* transgenic mouse line are already described in detail [17]. To induce *CreER^T2^*-mediated recombination of *Bax* in neural stem cells in the adult brain, mice of 11 weeks of age were given 55 mg/kg tamoxifen (20 mg mL^−1^, T-5648, Sigma, St-Louis, MO, USA) intraperitoneally, once a day for 5 consecutive days, or 10 mL/kg body weight of corn oil for vehicle-treated mice. Animals were group-housed and kept under standard laboratory conditions (12/12 h light-dark cycle with lights on at 8:30 p.m. and room temperature at 22 ± 2 °C), in enriched cages (46 × 29 × 25 cm, PAULA Ferplast, Castelgomberto, Italy) for 4 weeks starting from the first tamoxifen injection. Access to food and water was ad libitum. Mice were then divided into two groups depending on whether they had received the unpredictable chronic mild stress regimen (UCMS) or not (non-stressed: NS). Mice subjected to the UCMS regimen were housed in 24 × 11 × 12 cm cages without any environmental enrichment from week 5 to week 9, while NS mice remained in enriched cages. All procedures were compliant with Directive 2010/63/EU guidelines on animal ethics (referral 2019092017334223, approved by the ethical committee CEEvdl).

### 2.2. Experimental Design

Mice of 11 weeks of age received 5 consecutive injections of tamoxifen and were housed in an enriched environment for 4 weeks, and then half of the mice underwent the UCMS protocol for 4 additional weeks, while the other half stayed in enriched cages and were not stressed. Based on stress application and treatment, mice were divided into four groups, whether they underwent the UCMS regimen or not (UCMS: *N* = 43; NS: *N* = 40) and whether they received tamoxifen or vehicle treatment (tamoxifen: *N* = 43; vehicle: *N* = 40). After the UCMS regimen, all mice were tested for behaviors; however, we separated mice into two cohorts depending on which behavior was assessed. The first cohort (*N* = 49) was tested for anxiety-like behaviors as well as anhedonic and goal-motivated behaviors, and at the end of the experiment, mice were sacrificed to quantify hippocampal neurogenesis. Additionally, a second cohort (*N* = 34) was tested for stress-coping behaviors and cognition only (Figure 2a). We tested mice for cognitive impairments with a water maze learning task. Although learning tasks are known to be affected by hippocampal neurogenesis, it is also frequently observed that hippocampal neurogenesis is modified by learning; for this reason, we chose to test mice in the water maze paradigm in a second cohort where only the stress-coping behaviors were assessed before in order to avoid any side effect of a learning-induced increase in hippocampal neurogenesis on depressive-like and anxiety behaviors and/or on the quantification of hippocampal neurogenesis.

### 2.3. UCMS

The UCMS regimen was used as previously described [10,45]. Briefly, UCMS mice were isolated in individual cages and subjected to various socio-environmental stressors of mild intensity on a daily basis according to an unpredictable schedule for 4 weeks. Stressors included removal of sawdust, damping the sawdust, replacing the sawdust with water at 21 °C, repeated sawdust changes, tilting the cages at 45°, placing a mouse into a cage that had been occupied by another mouse, contention in small tubes, and alterations of the light/dark cycle.

### 2.4. Coat State

The coat state of each animal was assessed at the beginning of each week from week 5 to week 9 of the experiment in order to monitor the behavioral degradation induced by the UCMS regimen [10].

### 2.5. Nest-Building Test

Animals were isolated in bigger individual cages (Makrolon Type III, Tecniplast, Buguggiate, Italy) 12 h before the start of the test for habituation. One square piece of pressed cotton (5 × 5 cm) was placed in each cage at 7:30 a.m., and the quality of the nest was assessed at two time points: after 5 h and after 24 h according to an already described 1–5 rating protocol (Deacon, 2006). The mice were then put back in their home cages.

### 2.6. Light/Dark Box

The apparatus consisted of a lightbox with transparent sides (20 × 20 × 15 cm) and a dark box with the same dimensions and opaque sides. The boxes were connected by a small opaque plastic tunnel (5.5 × 6.5 × 10 cm). Animals were placed in the lightbox, and once the animal entered the tunnel, the test started and lasted for 5 min. The number of entries and the total time spent in the dark box were recorded. An entry into a box was recorded when the animal placed all four paws in the box. This procedure is based on the innate avoidance of mice for well-lit areas and is used as an indicator of anxiety-like behaviors [46].

### 2.7. Novelty-Suppressed Feeding Test (NSF)

As previously described [10], the apparatus consisted of a square (30 × 30 × 30 cm) with the floor covered with litter and illuminated by red light; a small food pellet was placed in the middle of the apparatus on a small piece of white paper (2 × 2 cm). This test is based on the natural aversion of mice for open areas, and they face a conflict between the drive to eat and the aversion for the open space. Once the mice were positioned in the apparatus (head facing the wall), the latency to explore and the latency to eat the pellet were measured. Once the mice started to bite the pellet, they were gently put back in their home cages with the pellet and allowed to eat for 5 additional minutes, and the consumption of the pellet was then recorded.

### 2.8. Splash Test

The test was conducted as previously described [47]. Mice were sprayed on their dorsal coats with a 10% sucrose solution in their home cages and under red light. Grooming behaviors were stimulated by the palatability of the solution. The duration of grooming was recorded during a period of 5 min after the spray.

### 2.9. Cookie Test

To test for anhedonic traits, mice were subjected to a reward maze test [11,48]. The apparatus was composed of three consecutive chambers (20 × 20 × 20 cm), which communicate through openings. During the test, mice were placed in the first chamber, and the palatable reward (butter cookie, Saint-Michel) was placed in the center of the third chamber. Once the mice entered the second chamber, the communication between the first and the second chamber was closed. The latency to eat the reward and the consumption of the reward was measured for up to 5 min from the moment the mice were placed in the first chamber.

On the testing day, common food pellets were removed from the cage lid 1 h before the test. Moreover, to minimize environmental neophobia, mice were habituated three times to the device, and the test was performed under red light. Mice were familiarized with the cookie by giving a sample every 2 days 1 week before the test (for more details, see Appendix A and methods).

### 2.10. Tail Suspension Test

Stress-coping behaviors were assessed by the tail suspension test: mice were suspended above the ground by their tails with tape for 6 min without any possibility of escape or hold onto any surface [49]. This test allows measuring the immobilization time, which reflects the resignation of the mice and then can illustrate depressive-like behaviors.

### 2.11. Flexibility/Inhibition in the Water Maze

This test aims to evaluate two aspects of executive functions: cognitive flexibility and inhibitory control. Executive functions represent a set of high-level cognitive processes that support the elaborations and control of complex and adaptive behavioral responses. Among these processes, cognitive flexibility represents the ability to switch between different strategies or behavioral responses, while inhibitory control represents the ability to inhibit or override a behavioral response previously learned that became inefficient or irrelevant in order to implement adapted goal-oriented strategies.

We used a cross-shaped water maze with 4 arms (N, E, S, W) placed in a circular pool (diameter: 90 cm). The water was maintained at a temperature of 22 ± 2 °C, and the light intensity at the center of the device was approximately 100 lux. Mice were trained to find a hidden platform placed in one of the maze’s arms: the platform (5 × 5 cm^2^) was placed at its extremity slightly below the water surface (between 1 and 1.5 cm). The arm N contained a visual cue (a card with a strong black-and-white contrast) and, just above its extremity, a small lamp lit the visual cue at 500 lux. Sensory cues were placed in the water: small plastic lenses to support contextual learning; whether the water was clean or filled with lenses indicated a different localization strategy to reach the platform. The departure occurred at the extremity of one of the 3 other arms (E, S, W), the mouse’s head angled toward the center of the maze, and the departure position varying from a trial to another.

Mice had to learn different strategies in order to find the platform depending on the context (with or without lenses). A specific context was then associated with a specific task: mice had to find the platform by allocentric navigation strategy depending on the location of the cue (“cue training”), or by egocentric navigation strategy using a specific sequence of directions, independently from the arm of departure (“direction training”). Accordingly, each mouse during the experiment was subjected to 2 different contexts, and each of these contexts was associated with a different task.

For the training paradigm, on days 1–2, mice had to learn one of the two strategies in the presence of the first tactile cue. They had to associate a specific context to a task (Context A-Task 1). To do so, each mouse was given 4 blocks composed of 5 trials in a day, with a 30-min inter-block delay and a 30-s inter-trials delay on the platform.

On days 3–4, mice had to learn the second strategy in the presence of the other tactile cue (Context B-Task 2). Similarly to the first 2 days, mice underwent a learning phase of 4 blocks of 5 trials each day.

For each trial, mice were released from different starting positions; if the mouse had not reached the platform within 1 min, the experimenter gently led the mouse to the platform.

Subsequently, 1 week after the training phase, mice underwent the flexibility test. Each mouse was given 6 blocks in a day, but this time we alternated the cue task and the direction task between blocks. Then, blocks 1-3-5 were in context A while blocks 2-4-6 were in context B; accordingly, mice had to alternate their strategies to find the platform (Figure 3b). For all trials, the latency to reach the platform and the total number of perseverative errors (perseverative/interfering errors were defined as entries in the arm where the platform should be located in the other learned task) were recorded.

Thereafter, to test for inhibition (reversal learning), each mouse was given 6 blocks in a day, and contexts that were previously associated with a specific task, “Context A-Task 1” and “Context B-Task 2” were switched into “Context A-Task 2” or Context B-Task 1”. Accordingly, mice had to inhibit the behavioral response previously learned to find the platform. For all trials, the latency to reach the platform and the total number of persistent errors (when mice enter the arm where the platform was located in the previously learned task) were recorded (for more details, see Appendix A and Methods).

### 2.12. Immunohistochemistry

At the end of the experiments, mice were injected with an overdose of pentobarbital solution (100 mg/kg, Dolethal^®^, Vetoquinol, Lure, France), then transcardially perfused with 50 mL of heparin saline solution to remove blood, followed by 100 mL of 4% paraformaldehyde (PFA) in phosphate buffer 0.1 M solution to fix the brain. Brains were then extracted and placed overnight in PFA 4% solution, then cryoprotected in sucrose solution (20%) and stored at 4 °C. For immunochemistry, brains were cut into 30-µm coronal sections with a cooled microtome (−20 °C, Leica CM 3050 S, Paris, France). In order to quantify the AHN, a free-floating immunochemistry against doublecortin (DCX), a marker of immature neurons, was performed. Briefly, a heat antigen retrieval in citrate buffer (10 mM, pH = 6) was performed on brain slices for 10 min at 95 °C followed by incubation with primary antibodies at 4 °C for 48 h (DCX antibody 1/750 dilution, ab18723; Abcam, Cambridge, UK) and incubation with secondary antibodies (Donkey anti-mouse Alexa Fluor555, 1/500 dilution, ab150106; Abcam) for 2 h at room temperature. Finally, slices were mounted onto slides, covered with Vectashield^®^ mounting medium (Vector Laboratories, Burlingame, CA, USA), and stored at 4 °C (for more details, see Appendix A and methods).

The immunolabeled sections were observed under a Zeiss Z.2 Imager microscope in emitted-light mode, and DCX-labeled cells were counted in the DG at X20 magnificence. An unbiased and blinded protocol was used to count the DCX-labeled cells in the granule cell layer of the DG along its septotemporal axis. For quantification, 7 matched sections were selected for each mouse (4 sections for the dorsal hippocampus from bregma −1.3 to −1.8 mm, and 3 sections for the ventral hippocampus from bregma −3.3 to −3.6 mm) and DCX cells were expressed as normalized cellular densities (DCX cells+/mm^2^). Additionally, to evaluate the maturation, DCX cells with at least tertiary dendrites were counted, the maturation index was then expressed as the ratio of DCX cells with at least tertiary dendrites over the total number of DCX cells.

### 2.13. Statistical Analysis

Data were analyzed using ordinary two-way ANOVAs or mixed-factor two-way ANOVAs (repeated measures over time) followed by a Fisher post hoc test when needed for multiple comparisons (only when *p* < 0.05). For behavioral tests and DCX quantification, we evaluated the effect of treatment (tamoxifen or vehicle) and stress (UCMS or NS); thus, we had two categorical factors. For the coat state, we had the stress and treatment as categorical factors, as well as the weeks. For the water maze task, we had stress, treatment, and blocks as categorical factors. All results are presented as mean ± SEM. Detailed statistical analyses are provided in the Appendix A.

## 3. Results

### 3.1. Genetic Deletion of Bax Gene Tends to Enhance Survival and Promotes Maturation in Adult-Born Neurons

We first analyzed the DCX-labeled cells in the total hippocampus (Figure 1a). We observed that UCMS did not impact the total DCX cells (F_1,12_ = 0.089, *p* = 0.771) or the maturation index (F_1,12_ = 1.180, *p* = 0.299) regardless of the treatment. Tamoxifen induced a main increase in the DCX-labeled cells (F_1,12_ = 4.755, *p* = 0.050) compared to the vehicle groups. Multiple comparisons showed no significant difference between non-stressed vehicle mice and non-stressed tamoxifen mice (Fisher: *p* = 0.236), but UCMS tamoxifen mice tended to have an increase in the DCX-labeled cells compared to UCMS vehicle mice (Fisher: *p* = 0,091). The effect of tamoxifen on the maturation index was more important (F_1,12_ = 19.322, *p* = 0.001), revealing an increase in the maturation in the non-stressed mice tamoxifen compared to vehicle mice (Fisher: *p* = 0.009) as well as in UCMS tamoxifen mice compared to UCMS vehicle mice (Fisher: *p* = 0.009). Additionally, we analyzed the DG separately along its dorso-ventral axis (Figure 1b–c).

Results showed that in the dorsal hippocampus, UCMS had the main effect by decreasing the total number of DCX-labeled cells (F_1,12_ = 4.968, *p* = 0.046), and we observed a trend toward a decrease in the DCX-labeled cells in UCMS vehicle mice compared to non-stressed vehicle (Fisher: *p* = 0.091) but no significant difference between non-stressed and UCMS tamoxifen mice (Fisher: *p* = 0.213). However, UCMS did not affect the maturation index (F_1,12_ = 2.260, *p* = 0.159). Tamoxifen only tended to increase the total number of DCX-labeled cells (F_1,12_ = 3.922, *p* = 0.071) but significantly increased the maturation index (F_1,12_ = 7.186, *p* = 0.020) in the dorsal hippocampus. Multiple comparisons revealed a trend toward an increase in the maturation index in the non-stressed tamoxifen mice compared to non-stressed vehicle mice (Fisher: *p* = 0.072) and in the UCMS tamoxifen mice compared to UCMS vehicle mice (Fisher: *p* = 0.094; Figure 1b).

Concerning the ventral hippocampus, no main effect of UCMS was observed in the total number of DCX-labeled cells (F_1,12_ = 2.942, *p* = 0.112) or the maturation index (F_1,12_ = 0.005, *p* = 0.946) compared to non-stressed group. However, we observed that tamoxifen tended to increase the total number of DCX cells (F_1,12_ = 3.948, *p* = 0.070) and significantly increased the maturation index (F_1,12_ = 40.715, *p* < 0.0001) in non-stressed tamoxifen mice (Fisher: *p* < 0.001) and UCMS tamoxifen mice (Fisher: *p* < 0.001; Figure 1c) compared to non-stressed vehicle mice and UCMS vehicle mice, respectively.

### 3.2. Depressive-Like and Anxiety-like Behaviors Induced by UCMS Were Partially Prevented by Increasing AHN

The effects of UCMS were first evaluated on physical assessments such as the coat state (Figure 2b) and assessed with the nest-building score, a measure of daily-living behaviors (Figure 2c). Two-way ANOVA with repeated measures revealed significant differences between groups for the coat state (week*UCMS*tamoxifen: F_5,225_ = 9.249, *p* < 0.0001). UCMS induced a gradual deterioration of the coat state (week*UCMS: F_5,225_ = 68.254, *p* < 0.0001), reaching statistical significance from the second week on (Figure 2b). Indeed, UCMS vehicle mice had a significantly increased score compared to non-stressed vehicle mice from week 2 to week 6 (week 2–week 6, Fisher: *p* < 0.00001), similarly, UCMS tamoxifen mice had a significantly increased score compared to non-stressed tamoxifen mice starting from the second week (week 2–week 6, Fisher: *p* < 0.001). Tamoxifen did not have effect on the coat state (F_1,45_ = 0.028, *p* < 0.968). Additionally, evaluation of the nest-building score highlighted a UCMS*tamoxifen interaction effect (F_1,45_ = 8.299, *p* = 0.006), showing a UCMS effect in vehicle mice, as UCMS mice had higher scores than non-stressed mice (Fisher: *p* < 0.0001). This UCMS effect was not present in tamoxifen mice: the nest-building score was not different between UCMS and non-stressed tamoxifen mice (Fisher: *p* = 0.905; Figure 2c). Moreover, non-stressed tamoxifen mice had a higher score than vehicle non-stressed mice (Fisher: *p* = 0.005). Interestingly, increasing hippocampal neurogenesis was not sufficient to prevent the coat state degradation induced by UCMS, but it appeared that it could affect daily-life measures such as nest-building.

The effects of UCMS on goal-directed behaviors were assessed by the duration of grooming following a splash of sucrose solution over the fur of the animal (Figure 2d). Our results revealed no effect of UCMS (F_1,45 =_ 1.604, *p* = 0.210) or tamoxifen (F_1,45_ = 0.792, *p* = 0.378).

Additionally, the cookie test was used to evaluate anhedonia (Figure 2e); to this end, we compared the latency to eat palatable food as well as the total consumption between groups. No difference between groups was observed for the consumption following UCMS (F_1,41_ = 0.856, *p* = 0.360) and tamoxifen injection (F_1,41_ = 0.043, *p* = 0.837). However, while UCMS had no significant effect on the latency to eat (F_1,41_ = 1.867, *p* = 0.179), an effect of tamoxifen was present (F_1,41_ = 5.593, *p* = 0.023), revealing a significant decrease in the latency to eat the cookie in UCMS tamoxifen mice compared to UCMS vehicle mice (Fisher: *p* = 0.008); this effect was not observed in non-stressed tamoxifen mice compared to non-stressed vehicle mice (Fisher: *p* = 0.498). Accordingly, even if UCMS did not elicit depressive-like behavior in the cookie test, it seems that tamoxifen-induced increased hippocampal neurogenesis was sufficient to affect motivated behaviors in UCMS mice.

In order to measure anxiety, we used different behavioral paradigms such as the novelty-suppressed feeding test (NSF) and the light/dark box (Figure 2f,g). We observed no effect of UCMS and tamoxifen in the NSF on the latency to smell and the consumption (UCMS: latency to smell F_1,45_ = 0.051, *p* = 0.823; consumption F_1,45_ = 0.293, *p* = 0.591; tamoxifen: latency to smell F_1,45_ = 1.716, *p* = 0.197; consumption F_1,45_ = 1.429, *p* = 0.238). Thus, because no effect of UCMS was observed, it renders the results non-conclusive on how neurogenesis could affect stress-elicited anxiety-like behaviors in the NFS test.

Anxiety was also assessed in the light/dark box, in which the frequency and time spent in the dark box were measured (Figure 2g). An effect of UCMS was observed for the entries in the dark box while no effect was seen for the time spent in the dark box (respectively: F_1,45_ = 14.269, *p* = 0.000; F_1,45_ = 0.970, *p* = 0.330). UCMS vehicle mice entered the dark box more compared to the non-stressed vehicle mice (Fisher: *p* = 0.003); similarly, UCMS tamoxifen mice entered the dark box more than non-stressed tamoxifen mice (Fisher: *p* = 0.037). Additionally, no effect of tamoxifen was observed in this test for the entries (F_1,45_ = 0.505, *p* = 0.481) or the duration spent in the dark box (F_1,45_ = 0.605, *p* = 0.441). Taken together, these results show that in our experiments, increasing hippocampal neurogenesis was not sufficient to prevent UCMS-induced anxiety-like effects.

In the second cohort, which later underwent the water maze paradigm, stress-coping behaviors were assessed in the tail suspension test (Figure 3a). Two-way ANOVA revealed only a trend regarding tamoxifen injection (F_1,30_ = 3.302, *p* = 0.079) but a main effect of UCMS (F_1,30_ = 7.039, *p* = 0.013). UCMS vehicle mice spent more time immobile compared to the non-stressed vehicle mice (Fisher: *p* = 0.015), while no difference was observed between UCMS tamoxifen mice and non-stressed tamoxifen mice (Fisher: *p* = 0.260). Moreover, UCMS tamoxifen mice spent less time immobile than UCMS vehicle mice (Fisher: *p* = 0.040). Our data revealed that while UCMS induced higher immobility, increasing hippocampal neurogenesis prevented the effects of UCMS in this paradigm, thus decreasing deficits in stress-coping behaviors.

### 3.3. AHN Promotion Partially Promotes Resilience to Stress-Induced Cognitive Impairments

Although depressive mood and anhedonia are considered core symptoms of depression, patients suffering from depressive episodes also exhibit important cognitive impairments [40]. As such, memory deficits, as well as difficulties in executive functions, are often observed. Because hippocampal neurogenesis has been widely linked to cognition [17,41,50,51], and that few studies have evaluated the protective effects of promoting hippocampal neurogenesis on stress-elicited cognitive impairments in an animal model of MD, we decided to evaluate the effect of increasing hippocampal neurogenesis on cognitive impairments induced by UCMS. Therefore, we used a water maze protocol in order to evaluate flexibility and inhibition (Figure 3b). Since associative learning can promote hippocampal neurogenesis [52], this cognitive task was performed on the second cohort after the TST in order to avoid any effect of learning-induced hippocampal neurogenesis stimulation on previous measures.

Our results showed significant differences between sessions in the flexibility task (Figure 3c), highlighting a progressive decrease in the latency to find the platform all along the procedure (session: F_5,135_ = 16.753, *p* < 0.0001); nevertheless, no effect of UCMS was observed (F_1,27_ = 0.001, *p* = 0.970). Moreover, UCMS elicited no difference for the average latency to find the platform (F_1,27_ = 0.060, *p* = 0.809). Tamoxifen did not induce a significant effect, but a trend was observed, showing a slight decrease in latency to reach the platform all along the procedure compared to vehicle mice (F_1,27_ = 3.072, *p* = 0.088) as well as a trend to decrease the average latency compared to vehicle mice (F_1,27_ = 3.267, *p* = 0.082). Moreover, while perseverative failures were not affected by UCMS (F_1,27_ = 0.008, *p* = 0.931), we observed a main effect of tamoxifen, which seemed to have induced a decrease in perseverative failures (F_1,27_ = 5.183, *p* = 0.031; Figure 3c). Multiple comparisons revealed no difference between non-stressed vehicle and non-stressed tamoxifen mice (Fisher: *p* = 0.162), whereas tamoxifen tended to decrease the perseverative failures in UCMS mice compared to UCMS vehicle mice (Fisher: *p* = 0.087).

For the inhibition task (Figure 3c), two-way ANOVA with repeated measures revealed significant differences among groups (session*UCMS*tamoxifen: F_5,135_ = 3.586, *p* = 0.004), showing a decrease in latency to find the platform all along the procedure for all groups (Session: F_5,135_ = 51,808, *p* < 0.0001) and a UCMS*tamoxifen interaction effect, revealing that UCMS vehicle mice spent more time to find the platform compared to all other groups (F_1,27_ = 9.152, *p* = 0.005). Indeed, in the two first blocks, UCMS vehicle mice spent more time to reach the platform compared to non-stressed vehicle mice (block 1, Fisher: *p* < 0,0001; block 2, *p* = 0.006) and the UCMS tamoxifen mice (block 1, Fisher: *p* < 0,0001; block 2, *p* = 0.014). As such, while UCMS seems to delay the learning of the new task in vehicle mice, increasing hippocampal neurogenesis seems to prevent this effect.

Additionally, two-way ANOVA revealed a main effect of UCMS (F_1,27_ = 5.514, *p* = 0.026) and tamoxifen (F_1,27_ = 6.432, *p* = 0.017) for the number of perseverative failures (Figure 3c) as well as a UCMS*tamoxifen interaction effect for the average latency to find the platform (F_1,27_ = 9.152, *p* = 0.005). UCMS vehicle mice had a higher number of perseverative failures compared to non-stressed vehicle mice (Fisher: *p* = 0.012); however, this UCMS effect was reversed in tamoxifen mice, as UCMS tamoxifen mice had a decreased number of perseverative failures compared to UCMS vehicle mice (Fisher: *p* = 0.006). Overall, UCMS vehicle mice spent more time to find the platform compared to non-stressed vehicle mice (Fisher: *p* < 0.001), an effect that was reversed in tamoxifen mice, as UCMS tamoxifen mice presented a decreased average latency compared to UCMS vehicle mice (Fisher: *p* < 0.001). Moreover, UCMS tamoxifen mice do not present any difference compared to non-stressed tamoxifen mice (Fisher: *p* = 0.642), leading to the idea that promoting hippocampal neurogenesis not only decreased the intensity of UCMS-induced cognitive deficits but was sufficient to protect mice from them.

These results indicate that increasing hippocampal neurogenesis before UCMS reverses the UCMS-induced reversal-learning impairment in the inhibition task. Moreover, while UCMS did not impact flexibility in our model, increased hippocampal neurogenesis affected the number of perseverative failures but not the time to reach the platform in the flexibility task.

## 4. Discussion

A majority of studies evaluated the impact of AHN on stress vulnerability through its ablation or the antidepressant effects of its increase on stress-induced depressive-like behaviors. However, it is still poorly described how promoting AHN could act as a factor of resilience. Additionally, whereas the involvement of AHN in cognitive functions is now well described, how promoting AHN could prevent stress-induced cognitive deficits remains less studied. In the present study, to explore the contribution of AHN in the resilience to chronic stress, we used transgenic mice in which AHN was selectively promoted, and we aimed to examine the impact of increasing AHN before the onset of stress in a well-characterized animal model of MD, the UCMS model. Our results showed that the UCMS regimen induced a depressive-like phenotype in vehicle mice, characterized by an altered coat state and nest-building score, changes in stress-coping strategies in the TST, and anxiety-like behaviors in the light/dark box as well as cognitive deficits in a water maze task. However, increasing AHN before the stress was sufficient to prevent the effect of stress in some depressive-like behaviors (TST, nest-building score) as well as in executive functions (reversal learning).

Exposing mice to a 4-week UCMS regimen reduced hippocampal neurogenesis in the dorsal hippocampus, and even if it did not reach significance, tamoxifen tended to reverse this reduction. This stress effect on hippocampal neurogenesis is consistent with previous results underlying the negative impact of stress on the survival of adult-born neurons [45,53,54]. However, it seems that the ventral hippocampus was not affected by UCMS in our study, possibly due to the fact that before the UCMS regimen, mice were housed in an enriched environment, which could have partially prevented the effects of stress [31]. Moreover, most studies that observed a UCMS-induced decrease in AHN used protocols of 6 to 8 weeks; it is then highly probable that our 4-week protocol was not sufficient to reach a significant decrease in AHN in the ventral hippocampus [11,45,55,56,57].

Nevertheless, tamoxifen injection significantly increased the number of adult-born neurons in the total hippocampus and the maturation of these adult-born neurons in the dorsal and the ventral hippocampus. Therefore, our results are in line with previous studies using this transgenic model, which found increased survival and maturation, but no effect on proliferation [17,45,57].

Adult-born neurons have distinct functions along the dorso-ventral axis of the hippocampus: while the dorsal population is mainly involved in cognitive functions, the ventral population is more important for mood and stress regulation [58,59,60]. Our model was able to enhance AHN all along the dorso-ventral axis, therefore allowing us to investigate the effects of promoting AHN on the development of a depressive-like phenotype but also on cognitive functions. However, because our transgenic model is based on the selective ablation of the pro-apoptotic Bax gene in Nestin-expressing stem cells, one could think that behavioral changes could not solely be the consequences of the increase in adult-born neurons but also of changes in the glial cells ratio. Indeed, it has been repeatedly found that MD was associated with glial loss [61], raising the possibility that affecting the ratio of glial cells could alter behaviors in our model. It is nevertheless unlikely, as previous characterizations of this transgenic line showed no difference in the proportion of neurons and glia [17,18]. This previous characterization led us to think that our results were mostly explained by the increase in survival and maturation of adult-born neurons.

### 4.1. Increasing Adult Hippocampal Neurogenesis Partially Promotes Resilience to the Depressive-Like Phenotype Induced by UCMS

Daily-life measures such as the coat state and the nest-building score revealed that UCMS induced a significant deterioration of the coat state starting one week after the beginning of the UCMS regimen, as well as changes in the nest-building score. These UCMS-induced changes were prevented only for the nest-building test, while degradation of the coat state was not reversed by increasing AHN.

Additionally, we observed UCMS-induced impairments of motivational behaviors in the TST illustrated by an increase in immobility in vehicle mice that was not observed in mice treated with tamoxifen. Thus, promoting AHN did not only decrease the effect of stress on stress-coping behaviors but prevented their appearance. Even though we did not observe any significant anhedonic effect of stress in the cookie test, increasing AHN induced a decreased latency to eat. Therefore, even if our protocol did not induce significant anhedonic traits, promoting AHN could potentially affect them. According to these results, the stimulation of AHN seemed to have partially increased the resilience to the effects of chronic stress, as observed on despair and nest-building in our behavioral paradigms, whereas it did not affect the coat state. We previously observed a similar effect of enhanced AHN on these stress-coping strategies with this transgenic strain under a UCMS regimen; nonetheless, AHN was in this case promoted after the onset of stress, thus counteracting the UCMS effects once they were established [45,57].

Increasing AHN did not induce any behavioral changes in the splash test and the NSF test. However, in these tests, because chronic stress did not yield effects, it is difficult to conclude about the protective effects of AHN. This lack of stress effect in some of our behavioral paradigms could probably be explained by the fact that we used a shorter UCMS protocol compared to previous studies that highlighted stress-induced behavioral changes [11,45,57] and that our protocol did not elicit a decrease in AHN in the ventral part of the hippocampus. Nonetheless, these findings point out the fact that AHN alone may not be sufficient to induce antidepressant or anxiolytic effects in control non-stressed mice, which is consistent with previous studies evaluating the effect of AHN enhancement in animal models of chronic stress or MD [17,18].

Thus, increased AHN does not act as an antidepressant *per se* but rather plays a key role in counteracting the disruption of neural circuits and the behavioral modifications induced by chronic stress. AHN is involved in the negative feedback from the hippocampus on the hypothalamo-pituitary-adrenal axis, which is impaired by chronic stress [11,39,62]; thus, promoting AHN could prevent the deleterious effects of stress, whereby protecting from the development of symptoms. Therefore, our protocol had one limitation as we did not evaluate the stress-induced modifications of the HPA axis activity. Indeed, measures of glucocorticoid levels would have given us insights to understand the mechanisms through which increasing AHN could prevent stress-induced behavioral changes. Even though the present findings reveal only partial effects, it appears that increasing hippocampal neurogenesis could at least partially prevent chronic stress-related effects on stress-coping behaviors and anhedonia. However, it seems that increasing AHN hardly affects anxiety-like behaviors.

### 4.2. Increasing Adult Hippocampal Neurogenesis Promotes Resilience to Stress on Executive Functions

In addition, we examined the effects of increasing AHN on cognitive performance in a modified water maze paradigm, allowing us to measure inhibition and flexibility capacities. During the experiment, mice went through a 4-day training stage in which they had to associate two contexts with two different strategies to find the platform, according to either the self-location (egocentric navigation strategy) or the spatial cues (allocentric navigation strategy), with no strategy alternation within each training day (Figure 3b). This was followed by a flexibility task in which mice were exposed alternatively to each context and therefore had to alternate the strategy to find the platform. Finally, during an inhibition task, the previous association context/strategy was inverted, and the mice had to inhibit their previous learning and learn the novel context/strategy association to find the platform successfully. This protocol was designed to evaluate (1) the capacity to switch between already learned rules (flexibility) and (2) the capacity of reversal learning (inhibition) and examine if these cognitive functions are distinctly modulated by stress and/or AHN.

We observed a significant effect of stress in the inhibition task. UCMS vehicle mice spent more time to reach the platform compared to non-stressed mice and exhibited a higher number of perseverative responses, showing a deficit in inhibiting previously learned associative rules and persisted in searching the platform in the wrong arm. In our model, UCMS seemed to partially alter cognitive functions, in line with what is often observed in clinical studies [40,63]. Interestingly, these were reversed by tamoxifen treatment: increasing AHN before chronic stress was thus sufficient to promote stress resilience in the inhibition task. Surprisingly, we did not find any effect of UCMS for the latency to find the platform in the flexibility task, and AHN increase only tended to decrease latency without reaching significance. Nevertheless, increasing AHN seemed to have partially improved the capacities to switch between rules, as was observed by a decrease in perseverative errors in the flexibility task.

Interestingly, while spatial memory performance positively correlates with the level of AHN [64,65,66], it appears that a learning task such as the water maze influences the level of AHN [67,68]. Therefore, we separated our mice into two cohorts: mice that went through the water maze were not used for other behavioral tests than the TST to avoid any side effects of learning-induced increased neurogenesis on other behavioral tasks. For the same reason, we did not measure the levels of AHN on these mice, as this learning protocol could have induced augmentation of AHN even in stressed mice, thus rendering the quantification non-conclusive. Previous findings highlighted a putative role of AHN in reversal learning, while ablation induced cognitive deficits [50,69], stimulation of AHN was able to restore allocentric navigation and contextual memory in aged mice in a water maze task [29]. In addition, we showed that stimulation of AHN could improve performance in an inhibition task impaired by chronic stress and promote the appropriate updating of the learning strategy leading to a decrease in perseverative errors in the water maze task.

One of the main functions that have been proposed for AHN is pattern separation [17,24], which is the capacity to disambiguate two similar inputs coming from the entorhinal cortex into two distinct contextual representations. It is usually proposed that the sparse coding of the hippocampus plays a crucial role in this function as it decreases the probability of interferences between two similar encoded contexts. Since adult-born neurons seem to play a critical role in the modulation of the overall activity of the hippocampus [26,70], it is possible that they reduce memory overlapping through their inhibition control, thereby improving cognitive flexibility and reversal learning [19,41]. Indeed, X-irradiation of AHN increases the activity of mature granule cells in the DG along with a decreased performance in a reversal-learning task [50]. Conversely, increasing AHN decreased the DG excitability [25], improved contextual discrimination [17], and facilitated the encoding of new memories [71]. Interestingly, whereas the effects of AHN on executive functions were mainly observed in non-stressed conditions in previous studies, our data highlighted that directly promoting AHN before the onset of stress was sufficient to prevent the deficits of reversal learning. Taken together, these findings suggest that increasing AHN could play an important role in resilience to stress by modulating the sparse hippocampal activity and so promoting cognitive functions directly supported by the hippocampus. Moreover, because the hippocampus projects to prefrontal regions involved in cognitive processes and that stress disrupt this pathway [72], modulating hippocampal activity could act indirectly on these projections, thereby positively shaping executive functions and stress resilience.

### 4.3. Conclusions

Our findings add to accumulative evidence of the link between AHN and MD and open a novel insight into the role of AHN in the pathophysiology of MD. Our study highlights commonly accepted results such as a negative impact of chronic stress on hippocampal neurogenesis and behaviors. However, we also point out a novel insight into the neuronal mechanism underlying resilience to stress, showing that increasing hippocampal neurogenesis before the onset of stress could partially alleviate the depressive-like phenotype as well as the cognitive impairments. However, from a perspective point of view, more specific characterization of neuronal activity would be an asset to better understand the mechanisms underpinning the behavioral changes that we observed. Because adult-born neurons have been linked to neuronal control of the hippocampus [25,26,27,70], it would be interesting to clarify their direct and indirect involvement in neuronal circuits during behavioral tasks affected by stress.

## Figures and Tables

**Figure 1 cells-10-00972-f001:**
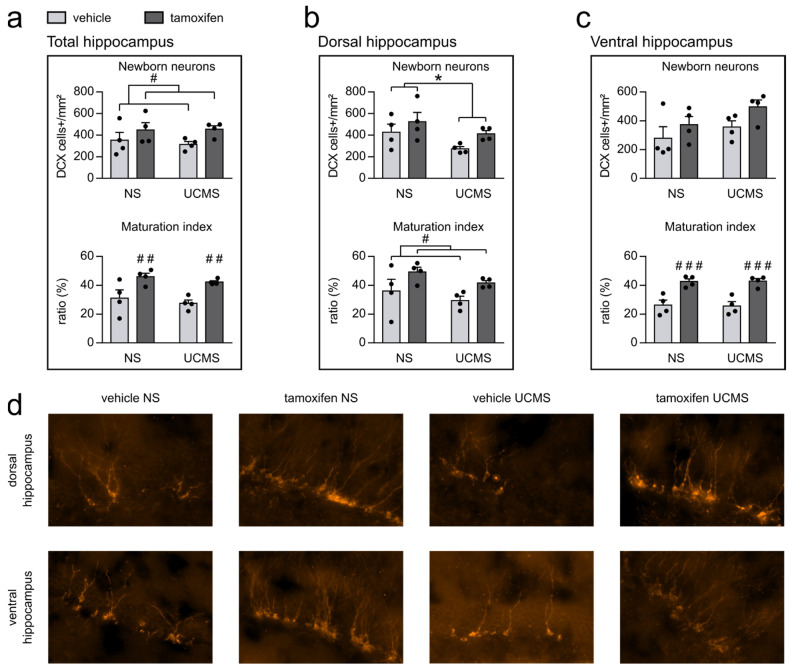
Modifications induced by UCMS and tamoxifen on hippocampal neurogenesis. (**a**) DCX-labeled cells and maturation index in the total hippocampus. UCMS did not affect the number of adult-born neurons or the ratio of DCX-labeled cells with at least tertiary dendrites on the total number of DCX-labeled cells, defined as the maturation index in the total hippocampus; however, tamoxifen increased both. (**b**) DCX-labeled cells and maturation index in the dorsal hippocampus. UCMS decreased the number of total DCX-labeled cells, and tamoxifen induced an increase in the maturation index. (**c**) DCX-labeled cells and maturation index in the ventral hippocampus. No effect of UCMS or tamoxifen was observed in the total DCX-labeled cells; however, tamoxifen significantly increased the maturation index. (**d**) Representative hippocampal sections immunostained for DCX from NS vehicle and tamoxifen mice, as well as UCMS vehicle and tamoxifen mice. ANOVA followed by post hoc multiple comparisons when necessary. *: *p* < 0.05 UCMS vs. NS; #: *p* < 0.05; ##: *p* < 0.01; ###: *p* < 0.001 vehicle vs. tamoxifen. UCMS: unpredictable chronic mild stress; NS: non-stressed.

**Figure 2 cells-10-00972-f002:**
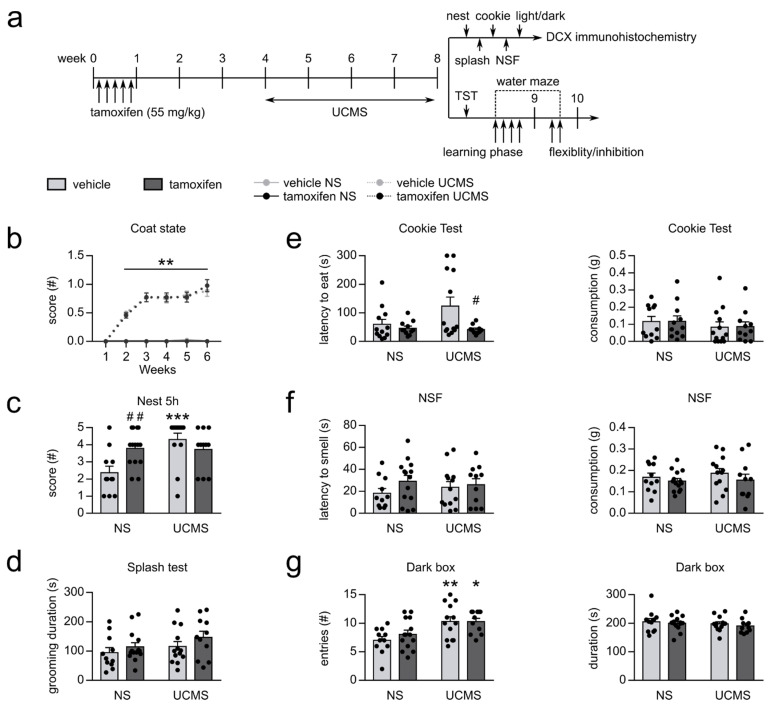
The effects of UCMS and tamoxifen were assessed in physical measures, motivational behaviors, anxiety-like and depressive-like behaviors in the first cohort. (**a**) Schematic representation of the experimental design. (**b–g**) UCMS elicited abnormal behaviors partially alleviated by tamoxifen. (**b**) UCMS induced deterioration of the coat state from the second week to the end of the experiment, which was not reversed by tamoxifen. (**c**) UCMS modified the nest-building score in vehicle mice but not in tamoxifen mice. (**d**) No effect of UCMS or tamoxifen was observed in the splash test. (**e**) No difference was found in the consumption of the cookie; however, UCMS tamoxifen mice had a decreased latency to eat the cookie compared to the UCMS vehicle mice. (**f**) No effect of UCMS or tamoxifen was observed in the NSF test. (**g**) In the light/dark box, UCMS mice had an increased number of entries in the dark box compared to non-stressed mice; nevertheless, no effect of UCMS and tamoxifen were seen for the time spent in the dark box. ANOVA, followed by post hoc multiple comparisons when necessary. *: *p* < 0.05; **: *p* < 0.01; ***: *p* < 0.001 UCMS vs. NS; #: *p* < 0.05; ##: *p* < 0.01 tamoxifen vs. vehicle. UCMS: unpredictable chronic mild stress; NS: non-stressed; NSF: novelty-suppressed feeding test.

**Figure 3 cells-10-00972-f003:**
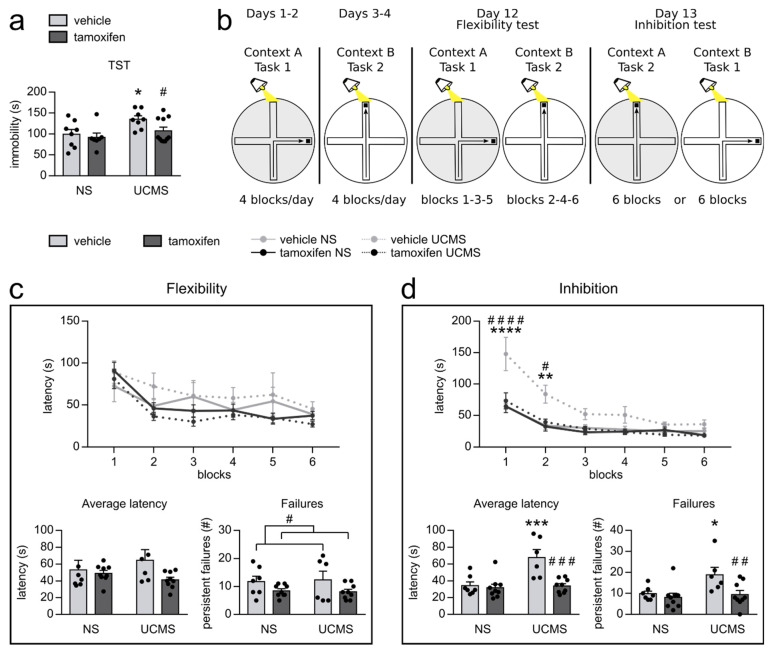
Promoting hippocampal neurogenesis counteracts UCMS-induced cognitive impairments in the water maze and TST in the second cohort. (**a**) The TST revealed that UCMS vehicle mice spent more time immobile than non-stressed vehicle mice, which was reversed by tamoxifen. (**b**) Schematic representation of the water maze protocol. While task 1 was based on an egocentric strategy (direction task), task 2 required an allocentric strategy (cue task). In context A (in gray), the water was filled with plastic lenses, whereas for context B (in white), the water was not. (**c**) A session effect was present, showing a decrease in the latency to reach the platform during the flexibility test. No effect of UCMS or tamoxifen was observed for flexibility, either in the evolution of the latency or the average latency; however, we observed that tamoxifen decreased the number of perseverative failures. (**d**) The inhibition test revealed a decrease in the time to reach the platform all along the test, but a UCMS*tamoxifen interaction effect highlighted the fact that UCMS vehicle mice spent more time to find the platform compared to all other groups in block 1 and block 2. Additionally, the comparison of the average latencies revealed that while UCMS elicited an increase in latency to reach the platform and an increase in perseverative failures, tamoxifen totally reversed these effects. (c-d) ANOVA, followed by post hoc multiple comparisons when necessary. *: *p* < 0.05; **: *p* < 0.01 UCMS vs. NS; #: *p* < 0.05; ##: *p* < 0.01; ###: *p* < 0.001; ####: *p* < 0.0001 tamoxifen vs. vehicle. UCMS: unpredictable chronic mild stress; NS: non-stressed; TST: tail suspension test.

## Data Availability

The data presented in this study are available on request from the corresponding author. The data are not publicly available due to privacy concerns.

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
