# Peer review of "Increasing Adult Hippocampal Neurogenesis Promotes Resilience in a Mouse Model of Depression"

_cells, 2021, doi:10.3390/cells10050972_

Round 1

Reviewer 1 Report

In this work, the authors used a mouse model to induce AHN increase before the exposure to UCMS. The aim was to evaluate the potential protective effect of AHN on stress, assessing for that depressive-like phenotype and executive functions. Results show that increasing hippo-campal neurogenesis before the onset of stress could partially alleviate the depressive-like phenotype as well as specific cognitive impairments.

The study is in general well performed and the topic is of high relevance to the field.

Main comments:

- The authors used an uCMS protocol for 4 weeks. Although they had an effect of uCMS in serveral behavioral paradigms tested, they failed to reach statistically significant stress effects on the reduction of hipocampal neurogenesis in the dorsal hippocampus, on the NSF, splash test, cookie test and behavioral flexibility in the water maze test. As such, regarding this specific analyses it is difficult to definitely conclude about the benefitial effects of increasing neurogenesis prior to the induction of uCMS (this should be taken in consideration while discussing the results). Possibly, the uCMS protocol should have been extended for at leats 2 additonal weeks to get more statiscally significant results regarding the parameters mentioned above. Still the results obtained regarding the effect of increasing AHN in the resilience to stress are of relevance to all other analyses performed.

- Still related to the point above the authors should provide corticosterone analyses of the mice after the 4 weeks uCMS protocol in animais untreated and treated with tamoxifen.

- the quality of the DCX staining in the images of the manuscript appear to be not optimal (probably due to the lower quality of the PDF provided). The authors should make check their quality.

Minor:

- something is missing in the following sentence in the first section of the results: “We first analyzed the DCX-labeled cells in the total hippocampus (Figure 1a) and we observed that while UCMS did not impact the total DCX cells (F 1,12 = 0,089, p = 0,771 ) or the maturation index (F 1,12 = 1,180, p = 0,299).”

- the NSF test is not described in the methods

- in the results part Fig 2f and g should be mentioned: “In order to measure anxiety, we used different behavioral paradigms such as the novelty suppressed feeding test (NSF) and the light/dark box (Figure 2f)”. – should be Fig. 2f and g

- please note that results of anxiety obtained with the NSF test are inconclusive in this study as there were no deficits observed in the uCMS group. Only results from the dark-box test allow to definitly conclude that increasing hippocampal neurogenesis was not sufficient to induce an anxiolytic-like effect.   

Reviewer 2 Report

1. It is necessary to more clearly formulate the novelty of the research in the abstract section, since the rationale is rather vague.
2. The introduction provides a good system of argumentation, however, some phrases need to be edited to make their meaning simpler and more understandable, so the introduction section needs to be stylistically edited.
3. Section 2.8 Materials and Methods of the "Cookie test " is not clearly described. It is necessary to clarify this section.
4. The Results section needs a complete revision. The authors cite a fairly large number of bihvioral tests for solving experimental problems, however, the description of the effects obtained in the course of experiments is minimized and requires clarification and clarification.
5. In the section Кesults, it is necessary to more clearly and in detail describe the changes in the experimental groups, in comparison with the control. This part of the work is very vulnerable and does not contain fundamentally new information in comparison with the already known data.
6. The work presents a very limited number of IHC markers to describe the complex and multicomponent process of adult neurogenesis. The IHC assessment is carried out in very general terms and does not take into account the complexity of this process. For a more detailed characterization, a wider set of molecular markers of adult neurogenesis is required. In particular, the ratio of proliferating and differentiated cells, the number of adult-type neuronal precursors, the process of glial differentiation, etc.
7. In the Discussion, their own results are poorly analyzed. This section is unnecessarily generalized and does not contain emphasis on our own results obtained in the course of numerous experiments. It is necessary to revise in detail the section "Results" and, on its basis, Discuss the situation in control and numerous behavioral data.

Round 2

Reviewer 2 Report

The authors have done a significant work to improve the text of the manuscript, which significantly influenced the quality of the work, improving it many times over. The authors took into account almost all the comments made during the primary review, and made the appropriate corrections to the new version of the text, as a result of which the quality of the article, in particular the quality of the results presented and their discussion, the scientific significance of the work and the interest for specialists, were greatly improved. I recommend a revised version of this article, after minor editorial changes, for publication in Cells.